# Extracellular Vesicles from *Lactobacillus rhamnosus BS-Pro-08*, Kefir Grain, Suppress Adipogenesis and Enhance Lipolysis in Adipocytes

**DOI:** 10.3390/ijms262311732

**Published:** 2025-12-04

**Authors:** Bi-Oh Park, Ho Woon Lee, Chang-Hyun Song, Miji Yeom, Seoungwoo Shin, Hyesoo Wang, Junbo Sim, Eunae Cho, Deokhoon Park, Eunsun Jung

**Affiliations:** Biospectrum Life Science Institute, Sinsu-ro, Suji-gu, Yongin-si 16827, Gyeonggi-do, Republic of Koreabioje@biospectrum.com (H.W.L.); biout@biospectrum.com (M.Y.); biost@biospectrum.com (S.S.); biocn@biospectrum.com (H.W.); bioba@biospectrum.com (J.S.); biozr@biospectrum.com (E.C.); pdh@biospectrum.com (D.P.)

**Keywords:** extracellular vesicles, *Lactobacillus rhamnosus*, adipogenesis, EV-enhanced lipolysis, postbiotic, adipose metabolism

## Abstract

Extracellular vesicles (EVs) derived from probiotic bacteria have recently emerged as postbiotic mediators that regulate host cellular responses. This study investigated the effects of EVs from *Lactobacillus rhamnosus BS-Pro-08*, isolated from kefir grains (Lacto EV), on adipocyte differentiation and lipid metabolism. Lacto EV treatment markedly suppressed the differentiation of 3T3-L1 preadipocytes into mature adipocytes, as reflected by reduced lipid accumulation and decreased expression of the adipogenic transcription factors peroxisome proliferator-activated receptor γ (PPARγ) and CCAAT/enhancer-binding protein α (C/EBPα). This inhibitory effect was most pronounced at the early stage of adipogenesis. In mature adipocytes, Lacto EV enhanced lipolysis in a dose-dependent manner, accompanied by increased glycerol release and total lipase activity. Interestingly, these lipolytic responses occurred despite reduced protein levels of adipose triglyceride lipase (ATGL) and hormone-sensitive lipase (HSL), suggesting that Lacto EVs may mediate an EV-enhanced lipolysis that is not fully explained by canonical ATGL/HSL signaling. Collectively, these findings demonstrate that Lacto EV modulates both adipogenic and lipolytic processes in vitro, providing insight into the metabolic actions of probiotic-derived vesicles.

## 1. Introduction

Adipose tissue serves as a central regulator of systemic energy homeostasis by balancing lipid storage and mobilization. This dynamic equilibrium is maintained through two opposing but coordinated processes: adipogenesis, the differentiation of preadipocytes into lipid-storing adipocytes, and lipolysis, the hydrolysis of triglycerides into free fatty acids and glycerol for energy utilization. Dysregulation of these processes leads to abnormal lipid accumulation and adipocyte hypertrophy, contributing to metabolic dysfunction associated with obesity and insulin resistance [1,2]. At the molecular level, adipocyte differentiation and lipid turnover are governed by a complex network of transcriptional and enzymatic regulators. Among them, peroxisome proliferator-activated receptor γ (PPARγ) and CCAAT/enhancer-binding protein α (C/EBPα) are master transcription factors that orchestrate the adipogenic gene expression program, whereas adipose triglyceride lipase (ATGL) and hormone-sensitive lipase (HSL) catalyze key steps of lipolysis [3,4]. Therefore, maintaining the proper balance between adipogenic activation and lipolytic signaling is critical for adipose tissue homeostasis and metabolic health [5,6,7].

Nanosized extracellular vesicles (EVs), typically 30–150 nm in diameter, mediate intercellular communication by transferring bioactive cargos such as proteins, lipids, and nucleic acids [8]. EVs participate in diverse physiological and pathological processes, including immune modulation, inflammation, and metabolic regulation. Among various sources, EVs derived from probiotic bacteria have gained growing interest for their ability to influence host cellular functions and metabolic homeostasis [9,10]. *Lactobacillus rhamnosus*, a well-characterized probiotic species, is widely recognized for its beneficial effects on gut health and immune regulation [11]. Recent research has extended this interest to the vesicles secreted by *L. rhamnosus*, which are increasingly considered bioactive mediators with functional relevance beyond the gut.

Several studies have demonstrated that *L. rhamnosus*-derived EVs can exert regulatory effects on host cells. For example, EVs from *L. rhamnosus* GG have been reported to modulate osteoblast–osteoclast balance and regulate bone microenvironment signaling in osteoporosis models, partly through the delivery of functional RNA cargos [12]. In addition, EVs derived from *L. rhamnosus* GG have been shown to carry bioactive molecules such as lipoteichoic acids and cell cycle-regulatory factors, which can influence immune signaling, cellular proliferation, and apoptosis in cancer-related settings [13]. These findings collectively support the concept that *L. rhamnosus* EVs function as postbiotic effectors capable of modulating host signaling pathways and cellular metabolism.

In parallel, emerging evidence suggests that probiotic-derived EVs may contribute more broadly to metabolic regulation. For instance, EVs from *Lactobacillus plantarum* were shown to alleviate diet-induced metabolic dysfunction in mice by improving hepatic lipid metabolism and enhancing AMPK signaling [14]. Together with other emerging reports, these findings suggest that probiotic EVs may broadly contribute to metabolic homeostasis. In this study, we hypothesized that EVs derived from *L. rhamnosus* may act directly on adipocytes to regulate adipogenesis and lipid turnover at the cellular level. To test this hypothesis, we investigated the effects of *L. rhamnosus BS-Pro-08*-derived EVs (hereafter referred to as Lacto EV) on adipogenic differentiation and lipolysis in 3T3-L1 adipocytes and explored their underlying molecular mechanisms through proteomic profiling.

## 2. Results

### 2.1. Identifying Extracellular Vesicles from Lactobacillus rhamnosus BS-Pro-08

We identified *Lactobacillus rhamnosus* from kefir grains and designated this strain as *L. rhamnosus BS-Pro-08*. *Lactobacillus* EVs (Lacto EVs) were isolated from culture supernatant using centrifugation, concentrated 15-fold with Amicon filters, and further purified by filtration. The isolated Lacto EV were characterized by nanoparticle tracking analysis (NTA) and transmission electron microscopy (TEM), based on the characterization criteria for extracellular vesicles established in previous studies [8,15]. NTA revealed that the particle size distribution ranged from 80 to 450 nm, with most particles exhibiting a size of approximately 106.2 nm and showed a concentration of 4.8 × 10^10^ particles/mL (Figure 1A). TEM analysis demonstrated that the Lacto EV possessed spherical, cup-shaped structures and were enclosed by lipid bilayers (Figure 1B). In addition, based on previous studies, the Gram-positive origin of the Lacto EV sample was verified by Western blot analysis of TSG101, outer membrane protein A (OMP-A), and lipoteichoic acid (LTA) [16]. The eukaryotic EV marker TSG101 and the Gram-negative marker OMP-A were not detected, whereas a distinct band was observed for the Gram-positive marker LTA, confirming that the Lacto EV were indeed Gram-positive-derived vesicles (Figure 1C).

Collectively, EVs were successfully isolated from *L. rhamnosus BS-Pro-08* and confirmed to be Gram-positive-derived extracellular vesicles with characteristic nanostructural features.

### 2.2. Lacto EV Inhibits Differentiation of Preadipocyte into Mature Adipocyte

The effects of Lacto EV on adipocyte differentiation were examined by seeding 3T3-L1 preadipocytes two days before confluence and inducing differentiation with a cocktail of dexamethasone, IBMX, and insulin, with or without Lacto EV. Medium was replaced every three days until full maturation (Figure 2A). To evaluate cytotoxicity, both preadipocytes and mature adipocytes were exposed to the specified concentrations of Lacto EV for 72 h. No significant change in cell viability was observed up to 1 × 10^8^ particles/mL in both preadipocyte and mature adipocyte stages (Figure 2B).

The anti-adipogenic effect of Lacto EV was assessed by treating differentiating cells every 3 days for 9 days. On day 9 of differentiation, morphological observations and Oil Red O staining demonstrated a concentration-dependent inhibition of intracellular lipid droplet formation. Quantification of Oil Red O staining confirmed a significant reduction in lipid content with increasing Lacto EV concentrations (Figure 2C). In addition, Lacto EV reduced triglyceride (TG) accumulation within adipocytes, as evidenced by a dose-dependent decrease in the activity of glycerol-3-phosphate dehydrogenase (GPDH), a crucial enzyme involved in TG synthesis (Figure 2D).

To elucidate the molecular mechanism mediating the anti-adipogenic effects of Lacto EV, we analyzed the expression of pivotal transcription factors that orchestrate adipocyte differentiation. These included PPARγ and C/EBPα, which are well-established master regulators that act in concert to promote the transcriptional program of adipogenesis. Quantitative PCR analysis revealed that Lacto EV treatment markedly downregulated the mRNA expression of both PPARγ and C/EBPα, with immunoblotting analysis confirming a similar reduction at the protein level (Figure 2E,F). Furthermore, the expression of fatty acid synthase (FASN) and fatty acid-binding protein 4 (FABP4), representative downstream targets of PPARγ and markers of lipid accumulation, was also significantly decreased in a dose-dependent manner (Figure 2E).

Taken together, these results suggest that Lacto EV impedes adipocyte differentiation by attenuating the transcriptional network that governs both the initiation and progression of adipogenesis.

### 2.3. Lacto EV Inhibits Adipogenesis at the Early Stage Through Downregulation of C/EBPα and PPARγ

To determine the specific stage at which Lacto EV exerts its anti-adipogenic effect, we assessed its impact during early and late phases of adipogenesis. Lacto EV treatment markedly suppressed adipogenesis when applied during the early phase, and this observation was consistent with lipid staining results. Under conditions a, b, e, and f, Oil Red O visualization demonstrated a substantial decrease in intracellular lipid accumulation. Lipid accumulation was markedly suppressed when Lacto EV was administered during the early stage of differentiation (conditions b, e, and f), compared to the non-early stage conditions (c, d, and g) (Figure 3A,B).

PPARγ and C/EBPα are well-established master regulators that initiate and coordinate the transcriptional cascade required for the commitment of preadipocytes to the adipocyte lineage [17,18]. Consistent with this, Lacto EV treatment under conditions a, b, e, and f markedly reduced the expression of both transcription factors at the mRNA and protein levels (Figure 3C,D).

These results indicate that Lacto EV inhibits lipid accumulation by modulating key adipogenic markers, particularly during the early stage of differentiation.

### 2.4. Lipolytic Activity of Lacto EV in Mature Adipocytes

To investigate the lipolytic effects of Lacto EV, 3T3-L1 cells underwent a 9-day differentiation process to generate mature adipocytes and were subsequently treated with the indicated concentrations of Lacto EV for 24 h. Glycerol release and lipase activity were then measured to assess lipolysis. As a result, glycerol levels in the culture medium increased, and lipase activity was also enhanced in a dose-dependent manner compared with the control group (Figure 4A,B).

Quantification of Oil Red O staining revealed that intracellular lipid content decreased by 8.72%, 19.82%, and 30.11% in a concentration-dependent manner. In addition, lipid droplet size was reduced with increasing concentrations of Lacto EV (Figure 4C).

These results confirm that Lacto EV enhances lipolysis in mature adipocytes.

## 3. Discussion

In this study, we investigated the potential of extracellular vesicles (EVs) derived from *Lactobacillus rhamnosus BS-Pro-08* (Lacto EV) on adipocyte differentiation and lipid metabolism. Our findings demonstrate that Lacto EV suppresses the early stages of adipogenesis by reducing the expression of C/EBPα and PPARγ, thereby inhibiting the differentiation of preadipocytes into mature adipocytes. Both C/EBPα and PPARγ are master transcriptional regulators that initiate and coordinate the gene expression program required for preadipocyte commitment and lipid accumulation, and their downregulation at this early stage is a critical intervention point to prevent subsequent adipose tissue expansion [19]. Early-stage inhibition is particularly important because it may limit clonal expansion and terminal differentiation, which are largely irreversible steps in adipocyte development [20]. The concurrent downregulation of downstream adipogenic markers such as fatty acid synthase (FASN) and fatty acid binding protein 4 (FABP4) further supports the inhibitory effect of Lacto EV on the adipogenic pathway.

In addition to its anti-adipogenic properties, Lacto EV enhanced lipolysis, as evidenced by a dose-dependent increase in glycerol release and lipase activity, indicating promotion of triglyceride breakdown in mature adipocytes (Figure 4). Lipolysis plays a central role in energy mobilization and is critical in maintaining metabolic flexibility, both of which are often impaired in obesity [21]. Enhanced lipolytic activity has also been associated with reduced adiposity [22]. Interestingly, despite these functional outcomes, the protein levels of endogenous lipolytic markers, including ATGL, total HSL, and phosphorylated HSL, were reduced (Appendix A). This finding suggests that Lacto EV may promote an EV-enhanced lipolytic response that is not fully explained by classical ATGL/HSL signaling and may involve alternative mechanisms.

Our proteomic analysis identified several EV-associated proteins that could contribute to these effects (Appendix A). Through the application of tandem mass tag (TMT) labeling and subsequent LC-MS/MS analysis, a total of 1228 proteins were identified. Based on their functional annotations and biological relevance to adipogenic regulation, 11 proteins were selected as potential anti-adipogenic candidates. In particular, the enrichment of GDSL-like lipase and α/β hydrolase proteins within EVs is consistent with the hypothesis that exogenous enzymatic activities may participate in triglyceride hydrolysis, potentially complementing or modulating host lipolytic pathways.

GDSL/SGNH hydrolases are characterized by a conserved GDS(L) motif and a canonical SGNH catalytic fold, exhibiting broad substrate specificity toward neutral lipids and phospholipids. These enzymes underpin microbial lipolytic activities, and bacterial lipases and phospholipases are known to remodel host lipid environments in various contexts, potentially affecting lipid homeostasis [23,24,25,26]. Similarly, α/β hydrolases comprise a broad class of esterases and lipases capable of hydrolyzing glycerides and reshaping host lipid pools [24,27]. Although we did not directly measure the enzymatic activities of these individual proteins in this study, their presence in Lacto EV provides a plausible molecular basis for the observed EV-enhanced lipolytic response in adipocytes.

This dual action—suppressing early adipogenic differentiation while promoting triglyceride breakdown in mature adipocytes—suggests that Lacto EV may influence adipocyte function through complementary molecular pathways. By concurrently attenuating transcriptional regulators of adipogenesis (PPARγ and C/EBPα) and engaging EV-associated hydrolases that could facilitate triglyceride hydrolysis, Lacto EV appears to modulate both lipid storage and mobilization within adipocytes. Such coordinated regulation is consistent with the concept that EVs act as multi-targeted mediators of metabolic signaling rather than acting through a single pathway.

Collectively, these findings indicate that functional proteins carried by Lacto EVs are likely involved in the regulation of adipogenesis and lipolysis, supporting the concept that bacterial EVs act as bioactive effectors capable of modulating host metabolic signaling. These observations align with emerging evidence that *Lactobacillus*-derived EVs can affect host metabolic processes [14]. Our results further expand this understanding by demonstrating the direct cellular actions of *L. rhamnosus* EVs on adipocyte differentiation and lipolysis.

Taken together, these findings suggest that probiotic-derived EVs such as Lacto EV may serve as bioactive modulators of adipocyte biology that are relevant to obesity and related metabolic disorders. A limitation of this study is that all experiments were conducted in 3T3-L1 adipocytes, and the in vivo relevance of Lacto EV therefore remains to be established. In addition, because our EV purification workflow did not incorporate cytoplasmic negative markers or density-gradient fractionation, further work implementing these approaches will be required to strengthen EV purity assessment in accordance with MISEV2018 recommendations. Further studies are warranted to identify the specific functional components within Lacto EV responsible for these effects and to evaluate their activities, safety, and mechanisms in appropriate in vivo models. Such investigations will be essential to determine whether these vesicles, or their key molecular constituents, can be developed into practical strategies for managing obesity-associated metabolic dysregulation.

## 4. Materials and Methods

### 4.1. Bacterial Strain Isolation

Kefir grains were washed with sterilized distilled water, suspended, and diluted in sterilized saline solution. The diluted samples were plated onto MRS agar (BD Difco, Franklin Lakes, NJ, USA) plates supplemented with 1% CaCO_3_ and incubated anaerobically at 37 °C for 48 h. Colonies producing yellow pigmentation and clear zones were presumed to be lactic acid bacteria and were subsequently isolated in pure culture. The strain was identified using 16S rRNA gene sequencing, and BLAST analysis (BLASTN 2.13.0+) determined the strain as *Lactobacillus rhamnosus*. The strain was designated as *L. rhamnosus BS-Pro-08*.

### 4.2. Preparation of Extracellular Vesicles and Characteristics

*L. rhamnosus BS-Pro-08* (accession number: KCTC 15876BP) was subcultured in a modified MRS medium (Biospectrum, Yongin, Republic of Korea) prepared from plant-based ingredients and then inoculated at 1% (*v*/*v*). The culture was incubated anaerobically at 37 °C for 60 h. The culture supernatant was harvested following centrifugation at 15,000× *g* and 4 °C for 25 min and filtered through a 0.45 µm membrane using vacuum filtration to remove large particles. A subsequent filtration with a 0.22 µm membrane was performed to eliminate residual bacteria. The resulting filtrate was concentrated 15-fold using an Amicon^®^ Ultra-15 system (100 kDa MWCO) (Millipore, Burlington, MA, USA) and washed with 0.1 µm filtered distilled water. To further remove residual soluble components and enrich EVs, the sample was then concentrated once more using the same Amicon^®^ Ultra-15 device (Millipore, Burlington, MA, USA). The purified *Lactobacillus*-derived extracellular vesicles (Lacto EV) were stored at −80 °C until use.

The concentration and size profile of the Lacto EV were assessed using nanoparticle tracking analysis (NTA) instrument (ZetaView PMX-120) (Particle Metrix, Inning am Ammersee, Germany). During measurement, optimal camera and laser alignment was ensured, and auto-symmetry adjustments were performed. The Lacto EV sample was serially diluted 1:800 in 0.1 µm filtered purified water to obtain approximately 150–200 particles per frame. The analysis was based on measurements from 11 positions conducted at 23.5 °C. Key instrumental parameters included a camera sensitivity of 80, a shutter speed of 100, 30 fps video capture (medium quality), and a particle detection area gated from 10 to 1000. All data was processed using the ZetaView software (Version 8.06.01 SP1).

For structural characterization, Lacto EV samples were prepared for transmission electron microscopy. Samples were applied to 300-mesh copper grids followed by negative staining with a 2% uranyl acetate solution. A JEM-2100Plus transmission electron microscope (JEOL Inc., Tokyo, Japan) operating at 200 kV was used to capture the electron micrographs.

### 4.3. Cell Culture and Differentiation

3T3-L1 cells were obtained from the American Type Culture Collection (ATCC, Manassas, VA, USA). The culture media components, including Dulbecco’s modified Eagle’s medium (DMEM), fetal bovine serum (FBS), bovine calf serum (BCS), and penicillin-streptomycin (PS), were acquired from Welgene (Welgene, Gyeongsan, Republic of Korea). Insulin, dexamethasone, 3-isobutyl-1-methylxanthine (IBMX) were obtained from Sigma-Aldrich (Sigma-Aldrich, Burlington, MA, USA). 3T3-L1 fibroblasts were maintained in DMEM supplemented with 10% BCS at a in a humidified 5% CO_2_ atmosphere at 37 °C. 3T3-L1 cells were seeded in 24-well plates at a density of 1 × 10^5^ cells per well and cultured until confluence, thereby inducing cell cycle arrest. Differentiation was induced with 500 μM IBMX, 2.5 μM dexamethasone, and 10 μg/mL insulin (DMI cocktail) in 10% FBS/DMEM for 3 days. The medium was then replaced with 10% FBS/DMEM containing 10 μg/mL insulin. On day 6, the cells were re-fed with 10% FBS/DMEM without insulin for 3 days. Unless otherwise stated, all in vitro experiments were conducted using Lacto EV at concentrations ranging from 0 to 1 × 10^8^ particles/mL. The specific concentrations applied in each experiment are indicated in the corresponding figure legends.

### 4.4. MTT Assay

Preadipocytic 3T3-L1 cells and mature adipocytes were exposed to Lacto EV for 72 h and then treated with 100 μg/mL MTT (Sigma-Aldrich, Burlington, MA, USA) in serum-free DMEM at 37 °C for 3 h. The media was discarded, and the accumulated formazan was solubilized by adding DMSO to the cells. Subsequently, absorbance was quantified at a wavelength of 570 nm using a multi-well plate reader.

### 4.5. Oil Red O Staining

Differentiated 3T3-L1 cells were washed once with DPBS (Welgene, Gyeongsan, Republic of Korea) and fixed with DPBS containing 4% formaldehyde for at least 30 min. Oil Red O Solution (Sigma-Aldrich, Burlington, MA, USA) was mixed with HPLC grade water at a 6:4 ratio to prepare the working solution, which was applied to fixed cells and stained for 45 min at ambient temperature. Once the staining step was completed, residual dye was removed by rinsing the cells five times with distilled water, after which the plates were allowed to dry completely and the bound dye was subsequently eluted with 100% 2-propanol. The absorbance was measured at 540 nm wavelength by using a multi-well plate reader.

### 4.6. Triglyceride (TG) Measurement

After discarding the medium, cells were rinsed twice with ice-cold DPBS and subsequently harvested in 25 mM Tris-HCl buffer (pH 7.5) containing 1mM EDTA. The pellet was sonicated for three cycles of 15 s at 20% amplitude using UP50H with MS7 (Hielscher ultrasonic GmbH, Teltow, Germany). The lysate was centrifuged at 10,000× *g* for 10 min prior to collecting the supernatant. TG content was measured using a TG colorimetric assay kit (Cayman Chemical, Ann Arbor, MI, USA), according to the manufacturer’s instructions.

### 4.7. GPDH Activity Assay

Cells were suspended in 25 mM Tris-HCl buffer (pH 7.5) with 1 mM EDTA and 1 mM β-mercaptoethanol. Cells were lysed by sonication and then centrifuged at 12,000× *g* for 20 min at 4 °C to obtain the supernatant, which was assayed using GPDH assay kit (TaKaRa Bio, Shiga, Japan).

### 4.8. Glycerol Release Assay

3T3-L1 adipocytes were fully differentiated by day 9 (D9) and subsequently incubated in serum-free medium for 16 h. Following overnight incubation, cells were treated with the indicated concentrations of Lacto EV for 24 h. The amount of free glycerol released into the culture supernatant was measured using a Glycerol Assay Kit (Abcam, Cambridge, UK) as directed by the kit manufacturer.

### 4.9. Lipase Activity Assay

Cells were rinsed twice with ice-cold DPBS, and lipase activity was assessed using the Lipase Activity Assay Kit (Cayman Chemical, Ann Arbor, MI, USA) according to the manufacturer’s protocol. Briefly, cells were harvested with 1× assay buffer and homogenized by sonication. After centrifugation at 10,000× *g* at 4 °C, the supernatant was used for the activity assay.

### 4.10. Quantitative Real-Time PCR (qRT-PCR)

To evaluate the mRNA expression level of *C/EBPα*, *PPARγ*, *FASN*, *FABP4* and *PPIA*, total RNA was prepared using the miniBEST Universal RNA Extraction Kit (TaKaRa Bio, Shiga, Japan). A total of 2 μg of the extracted RNA was used to synthesize cDNA using amfiRivert cDNA Synthesis Platinum Master Mix (GenDEPOT, Katy, TX, USA). qRT-PCR was performed using a CronoSTAR^TM^ (TaKaRa Bio, Shiga, Japan). Normalization was carried out using *PPIA* as the internal control. The primer sequences employed throughout this investigation are presented in Table 1.

### 4.11. Western Blot Analysis

Cell lysate was prepared with Cell Lysis Buffer (Cell Signaling Technology, Danvers, MA, USA) containing 1 mM PMSF. Equal amounts of protein (30 µg per lane) were subjected to SDS-PAGE and transferred to polyvinylidene difluoride (PVDF) membranes. Membranes were blocked in 5% non-fat dry milk for 1 h and then incubated overnight at 4 °C with primary antibodies against OMP-A, LTA (Invitrogen, Waltham, MA, USA), TSG101 (Abcam, Cambridge, UK), HSL, phospho-HSL, C/EBPα, PPARγ (Cell Signaling Technology, Danvers, MA, USA), ATGL, β-actin (Santa Cruz Biotechnology, Dallas, TX, USA). Afterwards, membranes were developed using HRP-conjugated IgG secondary antibody and HRP chemiluminescent substrates (Cell Signaling Technology, Danvers, MA, USA).

### 4.12. Proteomic Analysis

Proteomic profiling of EV was performed by Bertis Inc. (Gwacheon, Republic of Korea) using a Nano-LC Ultimate 3000 system/Thermo FAIMS Pro™ Orbitrap Exploris 480 mass spectrometer (Thermo Fisher Scientific, Waltham, MA, USA). Data-dependent acquisition performed and processed using SAGE software (version 0.14.7). Protein identification was carried out by matching the acquired mass spectra to *L. rhamnosus* protein sequences (Organism ID: 47715) in the UniProt database.

### 4.13. Statistical Analysis

All data are expressed as the mean ± SD or SEM, as indicated. Statistical analysis was performed using one-way ANOVA followed by Dunnett’s multiple comparisons test, and differences were considered statistically significant at *p* < 0.05.

## Figures and Tables

**Figure 1 ijms-26-11732-f001:**
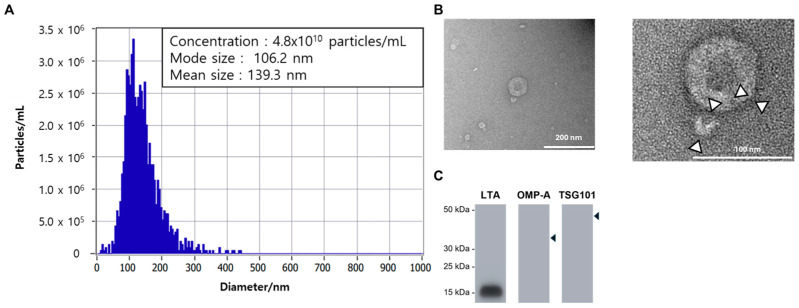
Identifying extracellular vesicles from *Lactobacillus rhamnosus BS-Pro-08*. (**A**) The isolated Lacto EV were characterized using nanoparticle tracking analysis (NTA). NTA revealed that the particle size distribution ranged from 80 to 450 nm, with the majority of particles exhibiting a size of approximately 106.2 nm. (**B**) TEM analysis demonstrated that the Lacto EV possessed spherical, cup-shaped structures and were enclosed by lipid bilayers. Arrows indicate representative vesicles. (**C**) Western blot analysis of Lacto EV to verify Gram-positive identity. The detection of TSG101 (eukaryotic EV marker), OMP-A (Gram-negative marker), and LTA (Gram-positive marker) was examined. Arrowheads indicate the predicted molecular sizes of the target proteins.

**Figure 2 ijms-26-11732-f002:**
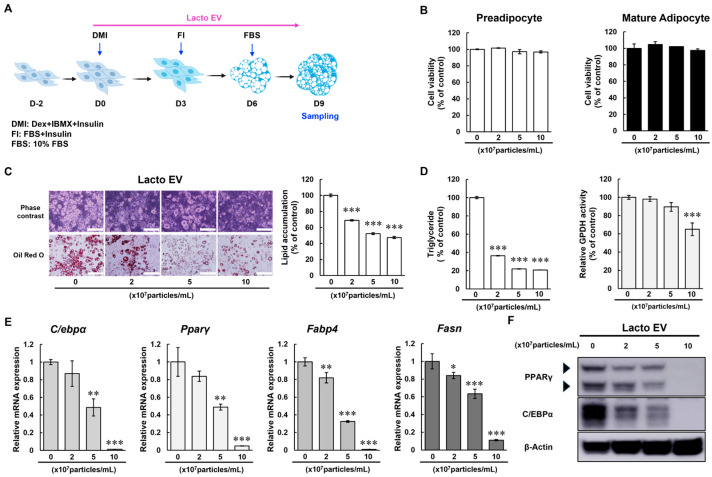
Lacto EV inhibits the differentiation of preadipocytes into mature adipocytes. (**A**) Schematic overview of experimental design. 3T3-L1 preadipocytes were seeded and cultured until 100% confluence (D0). Adipogenesis was induced with differentiation medium containing dexamethasone (Dex), 3-isobutyl-1-methylxanthine (IBMX), and insulin (DMI) for 3 days (D0–D3), followed by insulin-containing 10% FBS/DMEM medium (D3–D6), and then 10% FBS/DMEM without insulin until D9. Lacto EV was administered at each medium change. Cells were harvested at D9 for TG quantification, glycerol-3-phosphate dehydrogenase (GPDH) activity assay, qRT-PCR, and Western blot analysis. (**B**) Cell viability of preadipocytes and mature adipocytes after treatment with Lacto EV at the indicated concentrations (0, 2, 5 × 10^7^ and 1 × 10^8^ particles/mL) for 72 h. (**C**) Oil Red O staining of differentiated adipocytes treated with Lacto EV in a dose-dependent manner (scale bar, 100 µm). (**D**) Intracellular TG content and GPDH activity in differentiated adipocytes treated with Lacto EV. (**E**) mRNA expression levels of adipogenic marker genes (C/EBPα, PPARγ, FABP4, and FASN) determined by qRT-PCR. (**F**) Protein expression levels of C/EBPα and PPARγ determined by Western blotting; For PPARγ, the upper and lower arrowheads indicate PPARγ2 and PPARγ1, respectively. β-actin was used as a loading control. Data are expressed as means ± SD of at least three independent experiments. * *p* < 0.05, ** *p* < 0.01, *** *p* < 0.001 vs. control group.

**Figure 3 ijms-26-11732-f003:**
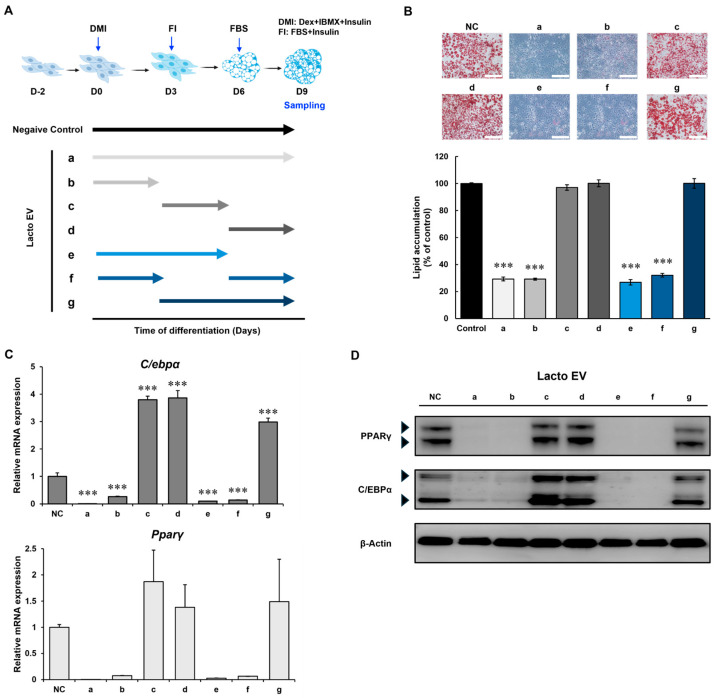
Lacto EV inhibits adipogenesis at the early stage. (**A**) Schematic diagram of the treatment conditions. Colored arrows indicate the periods during which cells were maintained in the corresponding media with Lacto EV (1 × 10^8^ particles/mL). (**B**) Oil Red O staining of adipocytes differentiated under the indicated conditions (scale bar, 100 µm). (**C**) qRT-PCR analysis of relative mRNA expression levels of C/EBPα and PPARγ. (**D**) Western blot analysis of changes in C/EBPα and PPARγ protein expression. For PPARγ, the upper and lower black arrowheads indicate PPARγ2 and PPARγ1, respectively. For C/EBPα, the upper and lower black arrowheads indicate p42 and p30, respectively. Data are expressed as means ± SD of at least three independent experiments. *** *p* < 0.001 vs. control group.

**Figure 4 ijms-26-11732-f004:**
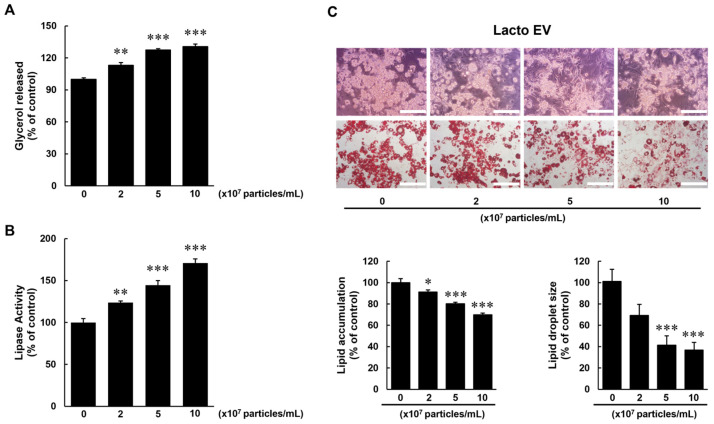
Lacto EV promotes lipolysis in adipocytes. (**A**) Glycerol release from differentiated adipocytes treated with Lacto EV at the indicated concentrations. Data are expressed as means ± SD (n = 3). (**B**) Differentiated adipocytes were assessed for lipase activity after treatment with Lacto EV at the indicated concentrations. Data are expressed as means ± SD (n = 3). (**C**) Analysis of lipid content and droplet size in mature adipocytes treated with Lacto EV at the indicated concentrations, as assessed by Oil Red O staining (scale bar, 100 µm). Lipid accumulation data are expressed as means ± SD (n = 3), and droplet size data are expressed as means ± SEM (n = 50). * *p* < 0.05, ** *p* < 0.01, *** *p* < 0.001 vs. control group.

**Table 1 ijms-26-11732-t001:** Sequences of primers used for qRT-PCR.

Genes	Primer	Sequence 5′ → 3′
** *C/EBPα* **	Forward	GCAAAGCCAAGAAGTCGGTGGA
	Reverse	CCTTCTGTTGCGTCTCCACGTT
** *PPARγ* **	Forward	ACATCAAGCCCTTTACCACA
	Reverse	CTGATGCTTTATCCCCACAG
** *FASN* **	Forward	CACAGTGCTCAAAGGACATGCC
	Reverse	CACCAGGTGTAGTGCCTTCCTC
** *FABP4* **	Forward	TGAAATCACCGCAGACGACAGG
	Reverse	GCTTGTCACCATCTCGTTTTCTC
** *PPIA* **	Forward	CAGGTCCATCTACGGAGAGA
	Reverse	CATCCAGCCATTCAGTCTTG

## Data Availability

The original contributions presented in this study are included in the article/Appendix A. Further inquiries can be directed to the corresponding author.

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
