# Peer review of "Extracellular Vesicles from Lactobacillus rhamnosus BS-Pro-08, Kefir Grain, Suppress Adipogenesis and Enhance Lipolysis in Adipocytes"

_ijms, 2025, doi:10.3390/ijms262311732_

Round 1
Reviewer 1 Report
Comments and Suggestions for Authors
This paper is well-structured, mechanistically interesting, and has potential novelty in linking probiotic-derived EVs with dual anti-adipogenic and non-canonical lipolytic actions. the logical progression is good; a few places need clearer comments around mechanism and more explicit explanation of experimental designs. Please refer to the attached file for details.

Comments on the Quality of English Language
Generally understandable but requires moderate editing for grammar, consistency, and EV terminology.
Author Response
Dear Editor and Reviewer 1,
We sincerely thank the reviewer for the thoughtful and constructive comments. We have carefully revised the manuscript to address all points, improve clarity, and strengthen our experimental framework. All revisions have been highlighted in the revised manuscript.
Below, we respond to Reviewer 1’s comments point by point.
Major Comment 1.
Mechanistic Basis for Non-Canonical Lipolysis
The claim that EVs promote triglyceride hydrolysis independently of the ATGL and HSL pathway is the central novelty of this work. The evidence provided does not yet demonstrate that Lacto EVs directly mediate lipolysis. Additional validation is necessary. The authors are encouraged to perform inhibitory experiments using known lipase blockers to determine whether the glycerol release induced by Lacto EVs is maintained. Direct enzymatic assays using purified EVs and lipid substrates would also clarify whether the vesicles possess intrinsic hydrolytic activity. If feasible, lipidomic profiling of intermediate species such as diacylglycerols and free fatty acids would strengthen the mechanistic argument.
Response
We appreciate this insightful comment regarding the mechanistic basis of the lipolytic effects of Lacto EV. In the original version, our description of “non-canonical lipolysis” could be interpreted as implying that we had definitively demonstrated EV-intrinsic triglyceride hydrolase activity independent of ATGL/HSL signaling. We agree that our current dataset is primarily correlative and does not yet establish such a direct and exclusive mechanism.
In the revised manuscript, we have therefore moderated our claims at multiple levels. First, we changed the title and Abstract to describe the phenomenon as “EV-enhanced lipolysis” rather than “non-canonical lipolysis” and now state that this response “is not fully explained by canonical ATGL/HSL signaling” (Title; Abstract, lines 24–29). Second, in the Discussion (lines 220–222, 237–239), we explicitly acknowledge that we did not directly measure the enzymatic activities of the GDSL-like lipases and α/β hydrolases identified by proteomics, and that their contribution provides a plausible, but not yet proven, molecular basis for the observed effects.
In addition, we now explicitly acknowledge this limitation in the Discussion and note that further studies will be required to identify specific functional components within Lacto EV and to evaluate their activities in appropriate in vivo models (lines 257–267).
Together, these changes clarify that our current study focuses on the cellular and proteomic correlates of EV-enhanced lipolysis and that a definitive demonstration of EV-intrinsic, ATGL/HSL-independent lipase activity remains beyond the scope of the present work.
Major comment 2.
Verification of Proteomic Candidates
The proteomic dataset suggests that several hydrolase-type enzymes are enriched in EVs, including GDSL-like lipases and α/β-hydrolases. These findings are interesting, but no follow-up confirmation is provided. It would be valuable to validate at least one candidate by Western blot or targeted mass spectrometry and to assess whether heat inactivation or antibody blocking of EVs alters their lipolytic effect.
Response
We agree that follow-up validation of the hydrolase-type proteins identified by proteomics would substantially strengthen the mechanistic conclusions. At present, our data show an enrichment of GDSL-like lipases and α/β hydrolases in Lacto EVs, but we did not directly assess the activity or functional necessity of individual candidates.
In the revised Discussion (lines 237–239), we now clearly state that “Although we did not directly measure the enzymatic activities of these individual proteins in this study, their presence in Lacto EV provides a plausible molecular basis for the observed EV-enhanced lipolytic response in adipocytes.”
In addition, the Discussion explicitly notes that further studies are required to identify specific functional components within Lacto EV and to evaluate their activities in more advanced in vitro and in vivo systems (lines 257–267). These statements clarify that the current proteomic findings are presented as mechanistic candidates rather than definitive proof, and that additional validation will be necessary in future work.
Major comment 3.
Characterization and Purity of EVs
The EV characterization uses nanoparticle tracking analysis, transmission electron microscopy, and a few marker proteins. The authors should describe in greater detail how potential contamination from bacterial debris or soluble proteins was excluded. The addition of negative markers, such as cytoplasmic proteins, or a density-gradient purification step would help satisfy MISEV2018 criteria and improve confidence in EV purity.
Response
We appreciate the reviewer’s valuable comments regarding EV characterization and purity assessment. In the revised manuscript, we have expanded the methodological description to clarify the rationale and steps implemented to minimize contamination and ensure EV integrity.
First, regarding the isolation workflow, we chose an ultrafiltration-based purification strategy rather than high-speed ultracentrifugation. Ultracentrifugation at >100,000 g is known to induce vesicle deformation, aggregation, and loss of functional cargo, whereas ultrafiltration applies substantially lower mechanical stress and is therefore more suitable for preserving the morphology and bioactivity of Gram-positive bacterial EVs. This approach is supported by recent studies demonstrating that ultrafiltration yields improved EV integrity and functionality compared with ultracentrifugation (Wadenpohl et al., Biotechnol. Bioeng., 2024 DOI: 10.1016/j.seppur.2023.126155). Furthermore, the use of ultrafiltration for microbial EV purification is consistent with recently proposed standardization recommendations for bacterial EV isolation (Choi et al., J Microbiol Biotechnol, 2025 DOI: 10.4014/jmb.2506.06011).
To further minimize contamination from soluble proteins, we performed two sequential rounds of ultrafiltration using the same Amicon® Ultra-15 device after membrane filtration, thereby enriching vesicles while reducing low-molecular-weight impurities. This additional concentration step has now been explicitly described in the revised Methods section (lines 281–283).
Second, we have clarified the analytical parameters used for particle quantification. In accordance with the reviewer’s suggestion, the revised manuscript now specifies that the Lacto EV sample was serially diluted 1:800 in 0.1 µm filtered water to obtain approximately 150–200 particles per frame for nanoparticle tracking analysis (NTA). This information ensures full transparency regarding particle number determination and dilution factors (lines 288–289).
As the reviewer noted, inclusion of additional negative markers (e.g., cytoplasmic proteins) or a density-gradient purification step would further strengthen alignment with MISEV2018 criteria. Although these experiments fall beyond the scope of the present study, we fully acknowledge this limitation and have added a statement in the Discussion indicating that future work will incorporate these approaches to advance EV purity assessment (lines 259–262).
We believe these clarifications and additions address the reviewer’s concerns while maintaining the integrity of the current experimental design.
Major comment 4.
Experimental Scope and Biological Relevance
The findings are limited to an in-vitro adipocyte model. While this model is suitable for mechanistic exploration, it does not demonstrate physiological relevance. The discussion should clearly acknowledge this limitation. If possible, the inclusion of preliminary in-vivo data, such as a short-term experiment in a high-fat-diet mouse model, would considerably enhance the manuscript.
Response
We appreciate the reviewer’s insightful comment regarding the experimental scope and biological relevance of our study. We fully agree that 3T3-L1 adipocytes provide a useful in vitro system for mechanistic investigation but do not establish physiological relevance at the whole-organism level.
In the present work, we did not perform in vivo experiments. Instead, we have now explicitly acknowledged this limitation and clarified the scope of our conclusions in the Discussion section. Specifically, we have added the following sentence toward the end of Section 3:
“A limitation of this study is that all experiments were conducted in 3T3-L1 adipocytes, and the in vivo relevance of Lacto EV therefore remains to be established.” (lines 257–259)
In addition, the subsequent sentence in the revised Discussion emphasizes the need for in vivo validation:
“Further studies are warranted to identify the specific functional components within Lacto EV responsible for these effects and to evaluate their activities, safety, and mechanisms in appropriate in vivo models.” (lines 262–264)
Together, these revisions clarify that our current conclusions are restricted to a cellular model and explicitly indicate that the physiological and therapeutic relevance of Lacto EV will need to be addressed in future in vivo studies, as the reviewer suggested.
Major comment 5.
Quantitative and Statistical Rigor
Several figures do not specify the number of biological replicates or the statistical methods used for multi-group comparisons. The methods section mentions only the Student’s t-test, which is insufficient for datasets with more than two conditions. The authors should indicate the number of independent experiments (n), the statistical tests applied, and how significance thresholds were defined.
Response
We thank the reviewer for highlighting the importance of statistical rigor and clarity. In response, we have revised both the Methods and the figure legends to explicitly state the statistical approaches used and the number of biological replicates.
Statistical methods (Methods 4.13)
In the original submission, the Statistical analysis section mentioned only the Student’s t-test, which indeed does not appropriately describe the multi-group comparisons used in this study.
We have now revised Section 4.13 as follows to reflect the actual statistical approach:
“All data are expressed as the mean ± SD or SEM, as indicated. Statistical analysis was performed using one-way ANOVA followed by Dunnett’s multiple comparisons test, and differences were considered statistically significant at P < 0.05.” (lines 385–387)
This revision clarifies that one-way ANOVA with Dunnett’s post hoc test was applied for datasets with more than two conditions.
The figure legends have been clarified to explicitly describe the number of independent experiments:
“Data are expressed as means ± SD of at least three independent experiments.” (lines 150–151, 177)
This wording makes clear that the bars and error bars in Figures 2 and 3 are based on 3 biological replicates.
Figure 4
For Figure 4, we additionally specified n values separately for each parameter and distinguished SD from SEM:
“(A,B) Data are expressed as means ± SD (n=3). (line 192, 194)
(C) Lipid accumulation data are expressed as means ± SD (n=3), and droplet size data are expressed as means ± SEM (n=50).” (lines 196–197)
This explicitly indicates how many independent experiments (or counted droplets) underlie each dataset and what type of variability measure is used.
Significance thresholds
Across Figures 2–4, the significance annotation is now consistently described in the legends as:
“*P < 0.05, **P < 0.01, ***P < 0.001 vs. control group.”
This addresses the request to clearly define the significance thresholds.
Alignment of significance annotations with ANOVA-based analysis
In line with the revised statistical description, the significance annotations in Figures 2–4 have been checked and updated so that all asterisks now explicitly correspond to the results of one-way ANOVA followed by Dunnett’s multiple comparisons test. Where necessary, individual significance markers were slightly adjusted to reflect the outcomes of the ANOVA/Dunnett analysis. These updates are reflected in the revised figure panels and legends.
Fig 2D Relative GPDH activity 5 x 107particles/mL * → ns
2E
Pparγ 5 x 107particles/mL * → **
Fabp4 2 x 107particles/mL * → **
Fasn 2 x 107particles/mL ns → * , 5 x 107particles/mL ** → ***
C/ebpα 5 x 107particles/mL *** → **
3C
C/ebpα b ** → ***
Pparγ *** → ns
4A
2 x 107particles/mL * → **
4B
2 x 107particles/mL *** → **
4C2
2 x 107particles/mL * → ns
Taken together, these revisions clarify the statistical tests applied, the number of independent biological replicates, and the definition of significance levels. They also ensure that all significance markers correspond to ANOVA followed by Dunnett’s multiple comparisons test, thereby improving the statistical rigor of the manuscript in line with the reviewer’s suggestion.
Major comment 6.
Terminology and Conceptual Consistency
The manuscript occasionally refers to bacterial EVs as “exosomes.” This terminology should be avoided. Consistent use of “extracellular vesicles(EVs)” would align the manuscript with accepted nomenclature and prevent confusion.
Response
We thank the reviewer for pointing out the importance of consistent and MISEV-compliant terminology.
As suggested, we have removed the term “exosome(s)” for bacterial vesicles throughout the manuscript and now consistently use “extracellular vesicles (EVs)” or “Lactobacillus-derived extracellular vesicles (Lacto EV).”
We believe these changes address the reviewer’s concern by aligning the manuscript with accepted EV nomenclature and removing potential confusion arising from the use of “exosome(s)” for bacterial vesicles.
Major Comment 7.
Interpretation of Findings
The authors frequently suggest therapeutic potential for obesity management. Such claims should be moderated, as the evidence currently supports only a cellular-level mechanism. The conclusion would benefit from focusing on the biochemical and molecular findings rather than potential clinical applications.
Response
We thank the reviewer for this important observation. We agree that the current study provides cellular-level mechanistic insights rather than direct evidence of therapeutic efficacy. Accordingly, we have removed or rephrased all statements that could imply clinical or anti-obesity applications and moderated our interpretation throughout the manuscript.
As noted in our response to Major Comment 4, the revised Discussion now clearly acknowledges that all findings are limited to an in vitro 3T3-L1 adipocyte model and do not establish organism-level metabolic outcomes. This limitation is explicitly described in the revised Discussion (lines 257–267), where we state:
“A limitation of this study is that all experiments were conducted in 3T3-L1 adipocytes, and the in vivo relevance of Lacto EV therefore remains to be established… Further studies are warranted… Such investigations will be essential to determine whether these vesicles… can be developed into practical strategies for managing obesity-associated metabolic dysregulation.”
These revisions ensure that the manuscript focuses on biochemical and molecular findings and avoids overinterpretation beyond the scope of the presented data.
Minor comment
Minor comment 1. Language Quality
The manuscript would benefit from professional English editing to correct grammatical inconsistencies and improve flow. Phrases such as “revelated” should be replaced with “revealed,” and “the cells were replaced” should be revised to “the medium was replaced.”
Response
Thank you for the suggestion. We carefully revised the entire manuscript to improve grammar, clarity, and readability. Spacing, phrasing, verb usage, and sentence flow were corrected throughout, and all identified language inconsistencies were addressed.
Minor comment 2. Figures and Documentation
Some Western blot images are faint or appear truncated. The uncropped versions included in the supplementary file should be explicitly cited in the main text, and molecular weight markers should be visible for reference.
Response
Thank you for pointing this out. We have now explicitly cited the uncropped western blot images for the main figures as well as those provided in the supplementary files.
Minor comment 3. Units and Presentation
Concentrations should be presented consistently using scientific notation, for instance 1 × 10⁸ particles per milliliter. The method used to determine EV particle numbers, including dilution factors and instrument parameters, should be clarified.
Response
We have revised the manuscript to present concentrations consistently using scientific notation (e.g., “1 × 10⁸ particles/mL,” “1 × 10⁵ cells/well”) throughout the text, figure legends, and Methods.
In addition, we expanded the NTA description in Section 4.2 (Preparation of extracellular vesicles and characteristics). The revised text now states that the Lacto EV sample was serially diluted 1:800 in 0.1 µm filtered purified water to obtain approximately 150–200 particles per frame and provides key ZetaView PMX-120 instrument parameters (camera sensitivity, shutter speed, frame rate, analysis positions, and particle detection range) (lines 290–297). This addresses the reviewer’s request regarding dilution factors and measurement settings for EV particle quantification.
Minor comment 4. Reference Style
References should follow a consistent format with uniform inclusion of journal names, volumes, and page numbers.
Response
We have reformatted the entire reference list to conform to the IJMS style. Journal titles, year, volume, and page (or article) numbers are now consistently included, and punctuation and abbreviation styles have been harmonized (e.g., use of full journal names as required by IJMS, consistent use of commas and semicolons, and inclusion of article numbers for MDPI and other journals that use them).
Minor comment 5. Data Availability
The statement that data are available upon request does not meet IJMS standards. Raw proteomic data should be deposited in a recognized public repository with accession information provided in the manuscript.
Response
We agree with this point. The raw proteomic LC–MS/MS data underlying the EV protein analysis have now been deposited in a public repository ProteomeXchange with identifier PXD065977.
The Data Availability Statement has been revised accordingly to indicate the repository name and accession number for the proteomic dataset, while clarifying that other processed data supporting the findings are available from the corresponding author upon reasonable request.
Minor comment 6. Typographical Corrections
Minor formatting inconsistencies, including spacing, hyphenation, and capitalization in figure legends, should be corrected.
Response
We have thoroughly reviewed and corrected typographical issues across the manuscript, including spacing, hyphenation, capitalization, and figure-legend formatting. These revisions have been applied consistently to ensure uniform presentation throughout the text.
Reviewer 2 Report
Comments and Suggestions for Authors
Park and co-authors isolated the EVs from Lactobacillus rhamnosus BS-Pro-08, Kefir grain, examined their bioactivities, as well as revealed their mechaniusm towards anti-adipogenic and lipolytic. The probiotics derived EVs are hot topics and the findings from the authors have a great significance for the obesity management. Nonetheless, the experimental design, methods, as well as statements of the presented work should be furtherly clarified and improved before its acceptance for publication.
- Would the author explain why the Lactobacillus rhamnosus BS-Pro-08, Kefir grain was selected? What is the rationale? The biofunctions of the Lactobacillus rhamnosus derived EVs should also been introduced. Please check the following references: 10.1016/j.compositesb.2023.111047; 10.1016/j.cej.2025.167807
- Would the author explain why they use ultra-filtration instead of ultra-centrifugation for the bacterial EV isolation? It is hard to separate the bacterial EVs with proteins using the ultra-filtration method. And the existing proteins from bacteria can also contribute to the anti-adipogenic and lipolytic effects as the authors have discussed (Figure S2). Please check the following reference: 10.26599/nr.2025.94908083
- For the concentration depended experiments, wide range from 105 to 109 particles/mL was typically used. It may not be reliable to examine the narrow range only within 107 particles/mL.
- The statement for bacterial extracellular vesicles should be unified. Either exosome or extracellular vesicles (EV) can be used. But the exosome vesicle is barely used for scientific statement.
- The scale bars were hard to see in Figure 2C, 3B, and 4B
- What were the differences between the upper and lower images in Figure 2C.
- The labels for the x-axis in Figure 2D were missing.
Author Response
Cover Letter: Response to Reviewer 2
Manuscript ID: IJMS-3998273
Title: Extracellular Vesicles from Lactobacillus rhamnosus BS-Pro-08, Kefir Grain, Ameliorate Obesity by Suppressing Adipogenesis and Promoting Non-Canonical Lipolysis in Adipocytes
Dear Editor and Reviewer 2,
We sincerely thank Reviewer 2 for the constructive and thoughtful comments provided. We carefully revised the manuscript in accordance with the suggestions. Below, we provide a detailed, point-by-point response. All modifications have been incorporated in the revised manuscript and highlighted for clear visibility.
Comment 1.
Would the author explain why the Lactobacillus rhamnosus BS-Pro-08, Kefir grain was selected? What is the rationale? The biofunctions of the Lactobacillus rhamnosus derived EVs should also been introduced. Please check the following references: 10.1016/j.compositesb.2023.111047; 10.1016/j.cej.2025.167807
Response
Thank you for the comment.
We selected Lactobacillus rhamnosus BS-Pro-08 based on preliminary internal screening results, in which EVs derived from this strain exhibited the strongest anti-adipogenic activity among multiple microbial candidates, including yeast-derived vesicles.
In addition, we have incorporated the two reviewer-recommended references in the Introduction ([12], [13] in the revised version), which provide broader context regarding the metabolic regulatory potential of Lactobacillus-derived EVs (lines 59–65).
Comment 2.
Would the author explain why they use ultra-filtration instead of ultra-centrifugation for the bacterial EV isolation? It is hard to separate the bacterial EVs with proteins using the ultra-filtration method. And the existing proteins from bacteria can also contribute to the anti-adipogenic and lipolytic effects as the authors have discussed (Figure S2). Please check the following reference: 10.26599/nr.2025.94908083
Response
We appreciate this important methodological point.
Our revised manuscript now provides a clearer rationale for selecting ultrafiltration over ultracentrifugation, supported by relevant published studies.
- Preservation of bacterial EV structure and bioactivity
Ultracentrifugation (>100,000 g) can induce vesicle deformation, aggregation, and reduced recovery, whereas ultrafiltration imposes lower mechanical stress and preserves vesicle morphology and biological activity more effectively.
Reference added:
Wadenpohl et al., Biotechnol. Bioeng., 2024 DOI: 10.1016/j.seppur.2023.126155 - Established use of ultrafiltration in microbial EV purification
Several recent reports have standardized ultrafiltration-based purification protocols for bacterial EVs, and our workflow aligns with these emerging best practices.
Reference added:
Choi et al., J Microbiol Biotechnol, 2025 DOI: 10.4014/jmb.2506.06011. - Additional clarification added to the manuscript
As requested, we have further strengthened the EV purification section by explicitly describing the additional steps included to reduce soluble contaminants, including:
- A second concentration step
“To further remove residual soluble components and enrich EVs, the sample was then concentrated once more using the same Amicon Ultra-15 device.” (lines 285–287).
Collectively, these clarifications address the reviewer’s concerns regarding EV purity while maintaining the core isolation strategy used in this study.
Comment 3.
For the concentration depended experiments, wide range from 105 to 109 particles/mL was typically used. It may not be reliable to examine the narrow range only within 107 particles/mL.
Response
Thank you for the observation.
Although the concentration range used in our study (2 × 10⁷ to 1 × 10⁸ particles/mL) is narrower than the full range reported in previous EV studies, this was an intentional experimental design.
Our preliminary screening demonstrated that this range produced consistent, measurable, and reproducible biological responses without inducing cytotoxicity. Because the aim of this work was to investigate mechanistic pathways (adipogenic transcription factors, lipase activity, proteomic signatures), we selected concentrations within the verified effective window, rather than conducting broad dose-response scanning.
We believe this focused concentration range supports the mechanistic objectives of the study while maintaining experimental reliability.
Comment 4.
The statement for bacterial extracellular vesicles should be unified. Either exosome or extracellular vesicles (EV) can be used. But the exosome vesicle is barely used for scientific statement.
Response
We agree with the reviewer. All terminology has now been standardized to “extracellular vesicles (EVs)” throughout the main text, figures, legends, and supplementary materials.
No instances of “exosome” remain in the revised manuscript.
Comment 5.
The scale bars were hard to see in Figure 2C, 3B, and 4B
Response
Thank you for pointing this out.
The visibility and contrast of scale bars in Figures 2C, 3B, and 4B have been corrected to ensure clear identification. In Figures 2C and 3B, the scale bar size was adjusted to 100 µm to improve readability, and this modification has been reflected in the corresponding figure legends (lines 147 and 173).
Comment 6.
What were the differences between the upper and lower images in Figure 2C.
Response
Thank you for raising this point.
We have now clarified in the revised Figure 2C that:
- Upper images: phase-contrast microscopy showing overall cell morphology
- Lower images: Oil Red O staining visualizing intracellular lipid accumulation
This distinction has now been made explicit to avoid confusion.
Comment 7.
The labels for the x-axis in Figure 2D were missing.
Response
Thank you for identifying this issue.
The x-axis labels (EV concentration categories) have now been added to Figure 2D in the revised manuscript.
Round 2
Reviewer 1 Report
Comments and Suggestions for Authors The paper has been revised to faithfully reflect the reviewer's comments. Comments on the Quality of English Language
Generally understandable but requires moderate editing for grammar, consistency, and EV terminology.
Reviewer 2 Report
Comments and Suggestions for Authors
All the concerned issues have been well addressed and the manuscript now is recommanded to accept.